# Roles of Tubulin Concentration during Prometaphase and Ran-GTP during Anaphase of *Caenorhabditis elegans* Meiosis

Ting Gong[1], Karen L McNally[1], Siri Konanoor[1], Alma Peraza[1], Cynthia Bailey[1], Stefanie Redemann[2], Francis J McNally[1]

In many animal species, the oocyte meiotic spindle, which is required for chromosome segregation, forms without centrosomes. In some systems, Ran-GEF on chromatin initiates spindle assembly. We found that in *Caenorhabditis elegans* oocytes, endogenously-tagged Ran-GEF dissociates from chromatin during spindle assembly but re-associates during meiotic anaphase. Meiotic spindle assembly occurred after auxin-induced degradation of Ran-GEF, but anaphase I was faster than controls and extrusion of the first polar body frequently failed. In search of a possible alternative pathway for spindle assembly, we found that soluble tubulin concentrates in the nuclear volume during germinal vesicle breakdown. We found that the concentration of soluble tubulin in the metaphase spindle region is enclosed by ER sheets which exclude cytoplasmic organelles including mitochondria and yolk granules. Measurement of the volume occupied by yolk granules and mitochondria indicated that volume exclusion would be sufficient to explain the concentration of tubulin in the spindle volume. We suggest that this concentration of soluble tubulin may be a redundant mechanism promoting spindle assembly near chromosomes.

## Introduction

Errors in chromosome segregation result in aneuploidy, a leading cause of embryonic lethality, and congenital defects if they occur during meiosis and cancer if they occur during mitosis (Nasmyth, 2002). Faithfull chromosome segregation in most eukaryotes relies on a bipolar spindle segregating chromosomes into daughter cells. The bipolar spindle is composed of thousands of microtubules, whose organization and stability are dynamically regulated to ensure proper chromosome attachment, alignment, and segregation (Kline-Smith & Walczak, 2004; Bennabi et al, 2016; Mullen et al, 2019).

In most mitotic cells, centrosomes at the two spindle poles act as major microtubule organization centers (MTOCs), in which spindle assembly factors (SAFs) are recruited to nucleate spindle microtubules (Petry, 2016; Prosser & Pelletier, 2017). Each centrosome contains a pair of centrioles and surrounding pericentriolar material (PCM) proteins (Kellogg et al, 1994; Bornens, 2012; Hinchcliffe, 2014; Wang et al, 2014; Sanchez & Feldman, 2017). However, mitotic cells lacking centrosomes can sometimes still assemble bipolar spindles, indicating the existence of additional pathways in spindle formation (Khodjakov et al, 2000; Conduit et al, 2015; Prosser & Pelletier, 2017). Moreover, centrosomes gradually degenerate during oogenesis, and female meiotic spindles in many animal species form without centrosomes (Heald et al, 1997; Schuh & Ellenberg, 2007; Dumont & Desai, 2012; Mikeladze-Dvali et al, 2012; Gruss, 2018). It is well-known that human oocytes, especially from individuals with advanced maternal ages or in vitro fertilizations are highly prone to meiotic spindle formation errors, resulting in aneuploid embryos (Angell, 1991; Thomas et al, 2021; Fair & Lonergan, 2023).

Two general pathways have been proposed to replace centrosomes and nucleate microtubules for spindle formation in oocytes, acentriolar cytoplasmic MTOCs and chromosome-directed spindle assembly (Li et al, 2006; Schuh & Ellenberg, 2007; Wu et al, 2022). In mouse oocytes, multiple de novo MTOCs originate from cytoplasmic microtubules before GVBD, which later increase in number and cluster into a multipolar spindle. These MTOCs lack centrioles but are enriched in PCM proteins (Schuh & Ellenberg, 2007). Cytoplasmic non-centrosomal MTOC-like structures enriched with different proteins have been observed in human oocytes, driven by microtubule-associated protein TACC3 (Wu et al, 2022). Non-centrosomal MTOCs have not been reported in oocytes of *Drosophila* or *Caenorhabditis elegans*.

Chromosome-directed spindle assembly has been studied extensively in *Xenopus* egg extracts where DNA-coated beads can direct bipolar spindle assembly (Heald et al, 1997). Four molecular mechanisms have been proposed to drive chromosome-directed spindle assembly: the Ran-GTP pathway, the Chromosome Passenger

[1]Department of Molecular and Cellular Biology, University of California, Davis, Davis, CA, USA  [2]Department of Cell Biology, University of Virginia, School of Medicine, Charlottesville, VA, USA

Correspondence: fjmcnally@ucdavis.edu

Complex (CPC) pathway, the kinetochore pathway, and the Augmin pathway. These mechanisms have been summarized in (Bennabi et al, 2016).

The small GTPase Ran has been demonstrated to play a critical role in chromosome-directed spindle formation in addition to its role in nuclear transport (Carazo-Salas et al, 1999; Kalab et al, 1999; Hetzer et al, 2002; Drutovic et al, 2020). The Ran GEF, RCC1, which creates Ran-GTP by promoting exchange of GTP for GDP, is localized in interphase nuclei and on condensed chromatin from prometaphase through anaphase during mitosis in cultured human cells (Ohtsubo et al, 1989; Moore et al, 2002), cultured rodent cells (Li et al, 2003), and *Xenopus* sperm chromatin incubated in M-phase *Xenopus* egg extract (Bilbao-Cortés et al, 2002; Li et al, 2003). Ran-GTP is thus concentrated in close proximity to chromosomes at Nuclear Envelope Breakdown (NEBD) where it can release inactive SAFs from binding with importins, thereby stimulating spindle assembly locally near chromosomes (Fig 1A). Cytoplasmic Ran-GAP converts Ran-GTP to Ran-GDP far from chromosomes, thus inhibiting spindle assembly in regions far from chromatin. In *Xenopus* egg extracts, Ran-GTP is necessary for spindle assembly around DNA-coated beads, constitutively active Ran-GTP stimulates spindle assembly in the absence of DNA-coated beads (Carazo-Salas et al, 1999) and RCC1 linked to beads is sufficient to drive bipolar spindle assembly (Halpin et al, 2011). The role of Ran-GTP in living oocytes has been less clear. An early study found that manipulating levels of Ran-GTP in mouse oocytes did not inhibit assembly of functional meiosis I spindles (Dumont et al, 2007). RCC1 is distributed throughout mitotic cytoplasm in *Drosophila* embryos (Frasch, 1991) and depletion of Ran-GTP did not abolish *Drosophila* meiotic spindle formation (Cesario & McKim, 2011). In contrast, Ran-GTP is required for meiotic spindle assembly in human oocytes (Holubcová et al, 2015). Later studies found that the requirement for Ran-GTP in mouse oocytes is redundant with pericentrin-dependent MTOCs (Baumann et al, 2017; So et al, 2022) whereas another study using a different method for inhibiting Ran-GTP found that it was required non-redundantly for meiotic spindle assembly in mouse oocytes (Drutovic et al, 2020). In *C. elegans*, depletion of Ran by RNAi prevented assembly of mitotic spindles (Askjaer et al, 2002) but did not prevent meiotic spindle assembly (Chuang et al, 2020).

The CPC is composed of Aurora B/C kinase, the inner centromeric protein (INCENP), Survivin, and Borealin, which target it to chromatin. The CPC is required for bipolar meiotic spindle assembly in mouse (Nguyen et al, 2018) and *C. elegans* (Divekar et al, 2021) although a cloud of disorganized microtubules is still nucleated near chromosomes. The Augmin complex is activated directly (Kraus et al, 2023) and indirectly (Petry et al, 2013) by Ran-GTP and recruits γ-tubulin to nucleate new microtubules on the sides of preexisting microtubules (Goshima et al, 2008; Lawo et al, 2009; Uehara et al, 2009; Petry et al, 2011; Colombié et al, 2013; Sánchez-Huertas & Lüders, 2015). However, no augmin homologs have been reported in the *C. elegans* genome.

A less-studied centrosome-independent pathway that may promote spindle assembly in the vicinity of chromosomes is the concentration of α/β-tubulin dimers in the nuclear volume at NEBD. Unpolymerized tubulin, monitored in cells treated with microtubule-depolymerizing drugs, is excluded from nuclei during interphase. During NEBD, rather than equilibrating to equal concentrations in the cytoplasm and nucleus, tubulin dimers have been reported to concentrate in the nuclear volume in *Drosophila* mitotic embryos (~1.6-fold, Yao et al, 2012), *C. elegans* mitotic embryos (~twofold, Hayashi et al, 2012), *Drosophila* S2 cells (~1.5-fold, Schweizer et al, 2015) and *Drosophila* neuroblasts (Métivier et al, 2021). The concentration of tubulin dimer independently of microtubule polymerization has been proposed to be related to binding to a spindle matrix, however, non-proteinaceous molecules like dextran (a polysaccharide) can also concentrate in the nuclear region (Yao et al, 2012), raising the question whether binding to a spindle matrix is necessary or not. Depletion of Ran by RNAi affects this concentration and causes spindle defects in *C. elegans* mitotic embryos (Hayashi et al, 2012) and *Drosophila* neuroblasts (Métivier et al, 2021). It is unclear whether the concentration of tubulin may be a component of the Ran pathway, or the defects observed might be the indirect result of altering the kinetics of nuclear import/export long before mitosis.

*C. elegans* female meiosis is unique in a few different ways. First, no distinct MTOCs containing PCM proteins have been observed in mature oocytes or meiotic embryos (McNally et al, 2006; Wolff et al, 2016). Second, RNAi knockdown of *C. elegans* Ran, RAN-1, decreases spindle microtubule levels but does not block meiotic spindle formation (Chuang et al, 2020). Third, *C. elegans* Katanin, composed of MEI-1 and MEI-2, is concentrated on chromosomes and is essential for formation of meiotic spindle poles (Srayko et al, 2000; McNally et al, 2014). Fourth, depletion of *C. elegans* γ-tubulin, TBG-1, by RNAi also leads to spindle microtubule loss but does not prevent meiotic division, although oocytes depleted of γ-tubulin and katanin by RNAi assemble extremely reduced levels of microtubules around chromosomes (McNally et al, 2006). Lastly, most SAFs remain cytoplasmic before GVBD (McNally et al, 2022). This suggests there might be novel mechanisms underlying the microtubule nucleation and spindle assembly without centrosomes in *C. elegans* oocytes. The mechanisms observed in *C. elegans* may be conserved in mammals and our studies may reveal factors that could be potential therapeutic targets to improve the efficacy of in vitro fertilizations.

## Results

### RAN-2[Ran-GAP] and RAN-3[Ran-GEF] Are Enriched on Anaphase Chromosomes

A previous study depleting Ran in *C. elegans* by *ran-1(RNAi)* showed female meiotic spindles still formed although with a reduced density of microtubules and showed relatively normal anaphase progression (Chuang et al, 2020). As RAN-1 may not have been fully depleted by RNAi, we created conditional knockdown worm strains of Ran-GEF and Ran-GAP by adding Auxin Induced Degron (AID) (Zhang et al, 2015) and HALO tag sequences to the endogenous *ran-2[Ran-GAP]* and *ran-3[Ran-GEF]* loci. RAN-2::AID::HALO and RAN-3::AID::HALO worms laid no eggs on auxin plates (Fig 1B). Previous depletion of RAN-3 or RAN-2 by RNAi caused >90% embryonic lethality but did not affect brood size (Askjaer et al, 2002). This suggests that

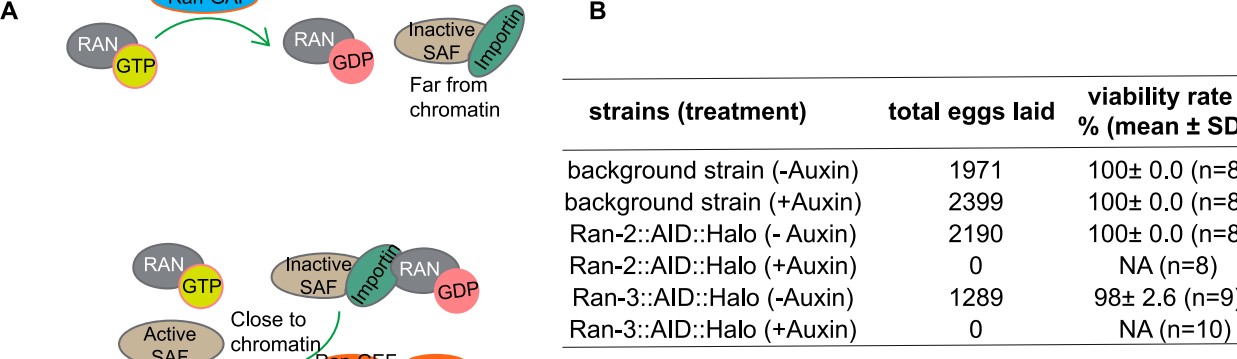

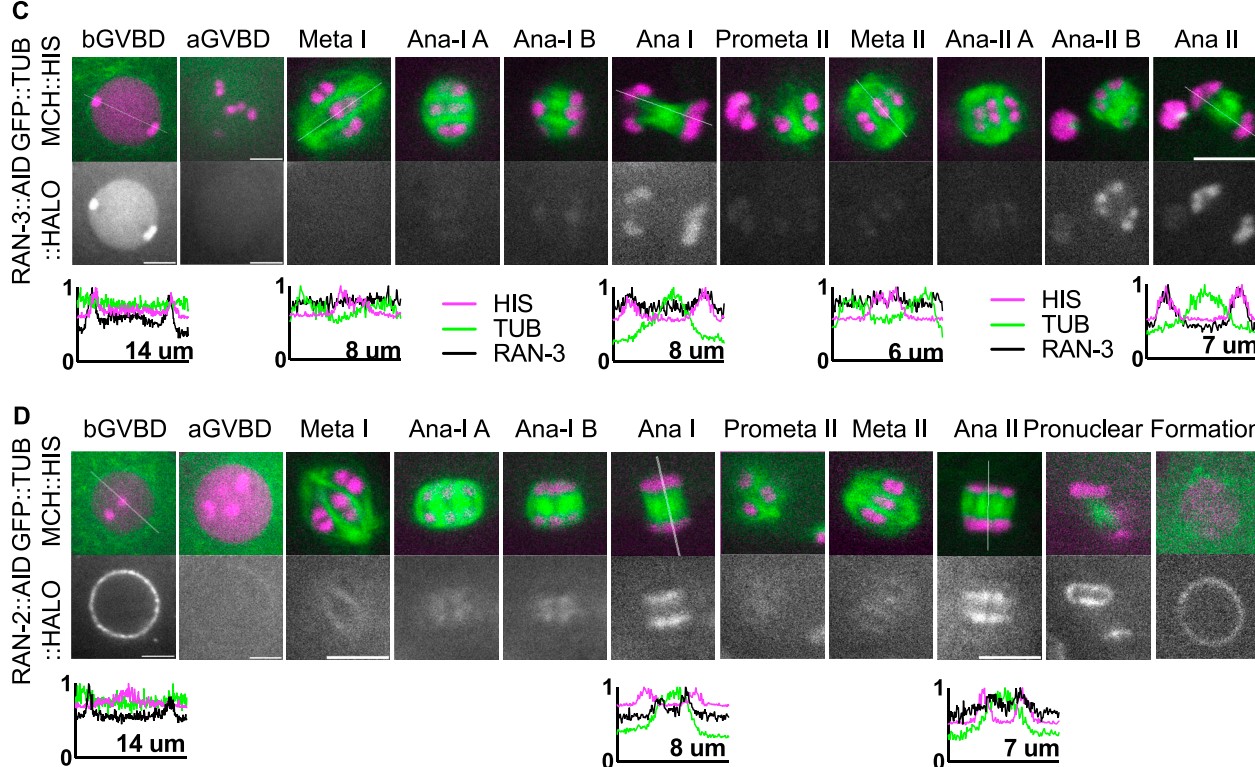

**Figure 1. RAN-2 and RAN-3 are associated with chromosomes during meiotic anaphase.**
**(A)** Diagram of Ran-GDP, Ran-GTP cycle activating spindle assembly factors near chromatin. **(B)** Embryonic viability of strains after depleting RAN-2 or RAN-3 by auxin-induced degradation. **(C)** Time lapse images of meiotic embryo expressing endogenously tagged RAN-3::AID::HALO, TIR1::mRuby, GFP::TUB (tubulin) and mCh::HIS (mCherry::histone H2b). **(D)** Time lapse images of meiotic embryo expressing endogenously tagged RAN-2::AID::HALO, TIR1::mRuby, GFP::TUB (tubulin) and mCh::HIS (mCherry::histone H2b). Graphs representing normalized fluorescence intensity along a 1-pixel-wide line scan (indicated by dashed line on the respective images) before GVBD, Ana I and Ana II are plotted on the bottom. Scale bars, 5 μm.

depletion of RAN-2 or RAN-3 through the AID system results in more complete depletion than RNAi.

It has been proposed that chromosome associated RCC1(Ran-GEF) generates Ran-GTP near chromatin whereas cytoplasmic Ran-GAP generates Ran-GDP far from chromosomes. Ran-GTP proximal to chromosomes releases inactive SAFs from binding with importins (Fig 1A). Consistent with these ideas, we observed RAN-3::AID::HALO in the nucleoplasm and chromosomes before GVBD (Fig 1C).

After GVBD, RAN-3 diffused from the nucleus and was not detected on chromosomes during spindle assembly in 12/12 embryos. RAN-3 only faintly associated with chromosomes at metaphase I and II. In contrast, RAN-3 strongly localized to chromosomes at anaphase I and anaphase II (Fig 1C). RAN-2::AID::HALO was strongly associated with the nuclear envelope before GVBD in 11/11 oocytes. Later, RAN-2 faintly labeled metaphase I and metaphase II spindles. At anaphase, RAN-2 localized to the spindle midzone, and its intensity

increased as anaphase progressed. It is strongly associated with the inner side of separating chromosomes (Fig 1D). These results suggest that RAN-3 and RAN-2 might function primarily at anaphase.

### Ran-GEF Is Required for Extrusion of the First Polar Body but not Metaphase I Spindle Formation

Ran regulators are also involved in nuclear transport, defects in which usually lead to small and leaky nuclei. By treating the worms with auxin for a brief period, we sought to only evaluate the function of RAN-3 and RAN-2 on meiotic spindle formation without disrupting meiotic prophase and nuclear transport. When treated with auxin for 4 h, expression of RAN-3::AID::HALO in the nucleoplasm and chromosomes in diakinesis oocytes was significantly reduced (Fig S1A and B). At 6-h auxin treatment, the expression level was comparable to control oocytes with no HALO expression. Moreover, −1 oocytes treated with auxin for 4 or 6 h were smaller than controls, whereas 36-h auxin resulted in much smaller nuclei (Fig S2A and B). Qualitatively, GFP:: tubulin partially leaked from the cytoplasm into nuclei after 4 h of auxin but GFP::tubulin leaking into the nuclei was much more severe at 36 h of auxin (Fig S2C). These results indicated that RAN-3 depletion after 4 h of auxin results in pre-NEBD defects and defects are more severe after 36 h of auxin. Expression of RAN-2::AID::HALO in the nucleoplasm and nuclear envelope in diakinesis oocytes was reduced to control levels after 4 h of auxin (Fig S1C and D). The sizes of oocyte nuclei were normal in RAN-2::AID::HALO worms after 4 or 6 h of auxin, and their nuclei were not leaky compared with controls (Fig S2A–C). RAN-2::AID:: HALO oocytes treated with auxin for 36 h were not quantified as these oocytes were severely disorganized. We therefore analyzed meiotic spindle assembly by time-lapse imaging after 4–6 h of auxin because these timepoints showed strong depletion but no detectable effect on nuclear function for RAN-2 and only moderate early effects for RAN-3.

Bipolar metaphase I spindles were observed in control worms (n = 15 no degron plus auxin; n = 10 *ran-3::AID* no auxin; n = 5 ran-2:: AID no auxin) (Fig 2A; Video 1). Consistent with the dispersal of RAN-3 and RAN-2 at NEBD (Fig 1C and D), 4–6-h auxin treatment of RAN-2:: AID::HALO (n = 33) or RAN-3::AID::HALO (n = 26) worms (Fig 2B and C) also resulted in bipolar metaphase I spindles. The mean metaphase I spindle GFP::tubulin pixel intensities were significantly decreased after RAN-3 depletion relative to both no degron controls or no auxin controls (Fig 2E and F). The length of metaphase I spindles was not significantly different than controls after depletion of RAN-2 or RAN-3 (Fig S1E) although a slight increase in metaphase II spindle length was observed after RAN-2 depletion (Fig S1F). The velocity of anaphase I (reported as the rate of increase in distance between separating chromosome masses) was significantly faster than no degron or no auxin controls after depletion of RAN-3 (Fig 2G). However, the velocity of anaphase II was not significantly different than controls (Fig S1G).

In 5/16 time-lapse sequences of RAN-3-depleted worms, chromosomes that moved toward the cortex at anaphase I merged into the metaphase II spindle. When viewed end-on, control metaphase II spindles have six univalents arranged in a pentagonal array (Fig 2A 810 s). In a side view of this pentagonal array during metaphase

or early anaphase, three chromosome pairs are in focus (Fig 2C 1,020 s). Merging of polar body-bound anaphase I chromosomes into the metaphase II spindle was inferred from more than six chromosomes in an end-on view (Fig 2B 1,060 s) or from direct observation when merging occurred in a favorable focal plane (Video 2). However, this frequency of polar body failure was not significantly different than the 1/10 failures observed in no auxin controls of the same strain (Fig 2D; *P* = 0.35 Fisher's exact test). Thus, failure to extrude the first polar body might be an artifact of time-lapse imaging or the genetic background. Conversely, the actual polar body failure rate might be much higher if the chromosomes that segregated toward the cortex at anaphase I do not move inward until later in development. To address both issues, we collected z-stacks of post-meiotic pronuclear to four-cell stage embryos dissected from worms at 5–7 h after placing worms on auxin. These embryos underwent meiosis 4–6 h after auxin treatment and the worms were not mounted for time-lapse imaging when these embryos underwent meiosis. 43/55 RAN-3 depleted embryos had a single polar body, whereas 29/29 no auxin controls of the same strain had two polar bodies (Fig 2D). This difference was significant (*P* = 0.0001 Fisher's exact test). Because the chondroitin layer of the eggshell is secreted during anaphase I (Olson et al, 2012), the first polar body is embedded in the eggshell and remains stationary at the anterior tip of the oval embryo, whereas the second polar body is inside the eggshell and moves to an internal location over time. Consistent with this, 18/19 no auxin control one-cell embryos had two polar bodies at the anterior tip whereas 19/20 no auxin control four-cell embryos had one polar body at the tip and one polar body internal. The single polar body of 14/15 RAN-3 depleted one-cell embryos was at the anterior tip whereas the single polar body of 15/15 RAN-3 depleted four-cell embryos was internal. The four-cell result was significantly different than a random distribution of 7/14 (*P* = 0.002 Fisher's exact test) and suggests that RAN-3 depletion causes resorption of chromosomes destined for the first polar body. Polar body extrusion defects after RAN-2 depletion compared with the same strain with no auxin were less definitive (*P* = 0.052 Fisher's exact test), and chromatin was extremely condensed in RAN-2 depleted post meiotic embryos, making it difficult to distinguish polar bodies from micronuclei at the cortex. Overall, these results suggest that the Ran-GTP pathway is important for limiting anaphase I velocity and polar body formation. Notably, RAN-3 is concentrated on chromosomes during anaphase and polar body extrusion (Fig 1). RAN-3 may have a more limited role in promoting the density of spindle microtubules at metaphase I when its localization on chromosomes is not discernible (Fig 1).

### Free Tubulin Is Concentrated in the Nuclear Volume at GVBD

At the onset of mitosis in *C. elegans* and *Drosophila*, soluble tubulin concentrates in the nuclear/spindle volume relative to the surrounding cytoplasm (Hayashi et al, 2012; Yao et al, 2012; Schweizer et al, 2015; Métivier et al, 2021). Because of microtubule polymerization is concentration dependent, this concentration might facilitate spindle formation in the vicinity of chromosomes in meiotic oocytes. In DMSO-treated control −1 oocytes, mNG::TBB-2 (β-tubulin) labeled microtubules in the cytoplasm and was excluded

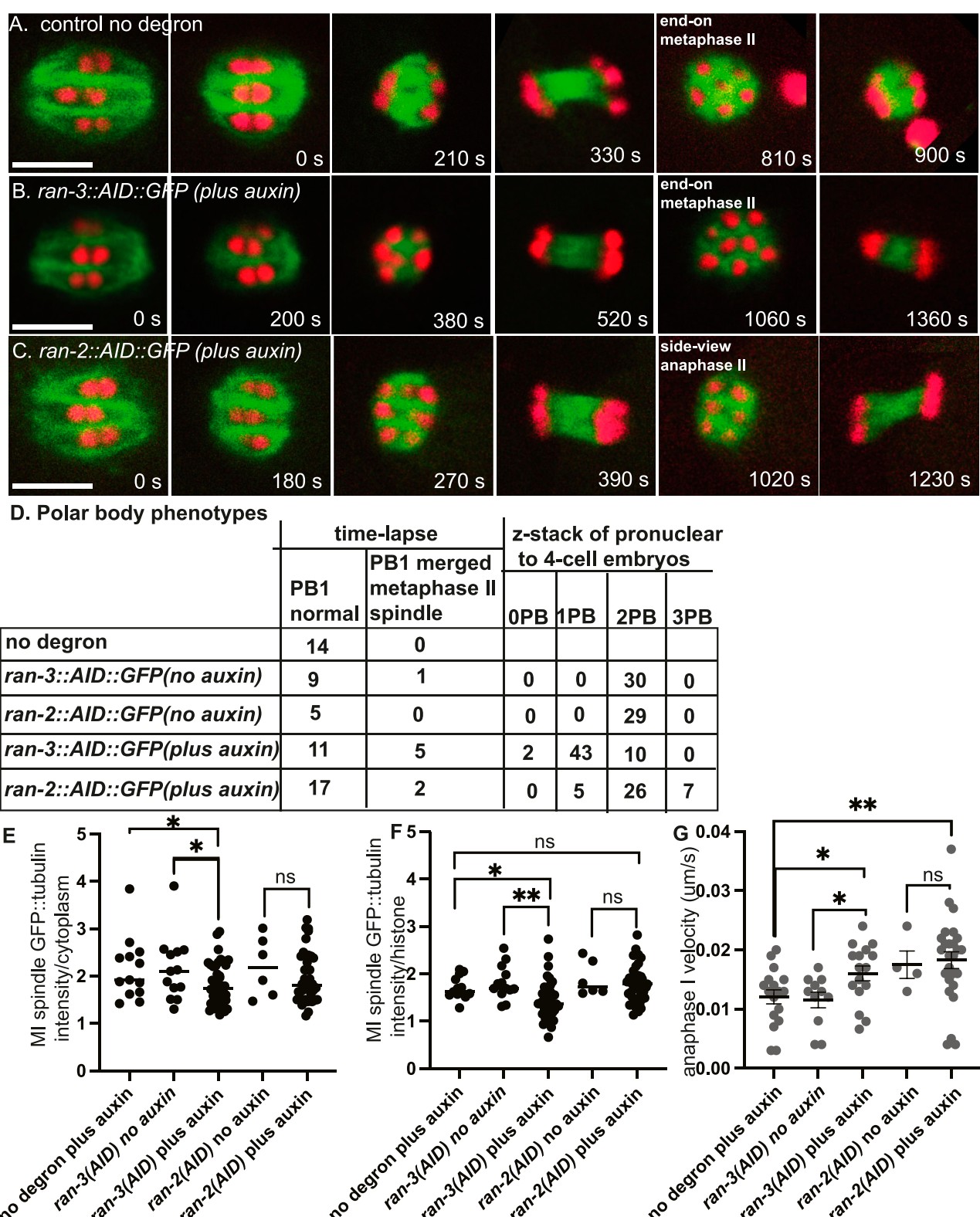

**Figure 2. Ran-GEF is required for extrusion of the first polar body but not for meiotic spindle formation.**
**(A, B, C)** Representative images from single focal plane, in utero time-lapse imaging from metaphase I through anaphase II of worms treated with auxin for 4–6 h. Metaphase I spindle morphology was not affected by depletion of RAN-2 or RAN-3. **(A)** At 810 s, 6 chromosomes are visible in an end-on view of a metaphase II control spindle. **(B)** At 1,060 s, eight chromosomes are visible in an end-on view of a RAN-3-depleted metaphase II spindle, indicating failure of the first polar body. **(C)** At 1,020 s, three chromosomes are visible in a side-on view of an early anaphase II RAN-2-depleted spindle. **(D)** Number of embryos with different polar body phenotypes

from the nucleus (Fig 3A and B). Upon fenestration of the nuclear envelope (GVBD: germinal vesical breakdown), indicated by leakage of non-chromosomal mCherry::histone out of the nucleus, mNG::TBB-2 fluorescence increased within the nuclear volume in 10/10 time-lapse sequences (Fig 3B) as previously described (McNally et al, 2006; Mullen & Wignall, 2017). Tubulin fluorescence then transformed into a "microtubule cage" (Mullen & Wignall, 2017) and eventually a bipolar spindle in 10 out of 10 time-lapse sequences. In oocytes treated with nocodazole to depolymerize microtubules, tubulin was diffuse in the cytoplasm before GVBD (Fig 3C), and still concentrated in the nuclear volume during and after GVBD (Fig 3C–E). Simple diffusion from the cytoplasm into the nuclear volume should result in equal fluorescence intensities of mNG::TBB-2 in the nucleus and cytoplasm but fluorescence instead increased in the nuclear volume to 1.2-fold greater than the cytoplasm (Fig 3E). After ovulation, chromosomes in nocodazole-treated zygotes were dispersed and loosely wrapped by sparse short microtubules. No spindle formation or chromosome separation was observed before pronucleus formation in 12 out of 12 time-lapse sequences.

We noticed that before GVBD, fluorescence of mNG::TBB-2 was detectable in the nucleus, which contradicts with the general assumption that the nucleus is void of tubulin (Fig 3B top panel, Fig S2B, average of the ratio of tubulin fluorescence in the nucleus to the cytoplasm: 0.69). To determine whether there is tubulin in the nucleus or if this is because of pinhole crosstalk from the spinning disk confocal, we captured images on a Zeiss laser scanning confocal microscope which has a single pinhole and therefore removes out of focus light more efficiently. The ratio of tubulin fluorescence in the nucleus to the cytoplasm before GVBD was significantly reduced to 0.33 (Fig S3A and B) in images from the Zeiss LSM compared with 0.69 from the spinning disk confocal. The difference in nuclear versus cytoplasmic fluorescence was also greater on the Zeiss LSM for other probes (Fig S3C and D). This suggests that the actual concentration difference between nucleus and cytoplasm are likely greater than the ratios reported from spinning disk confocal images.

Similar results in previous studies of mitosis led to the interpretation that alpha/beta tubulin dimers concentrate in the nuclear volume of unperturbed cells during spindle assembly. However, it is possible that nocodazole does not completely block microtubule polymerization and the fluorescence accumulating in the nuclear volume of nocodazole-treated oocytes represents accumulation of short microtubules. It is also possible that this phenomenon is induced by nocodazole and does not occur in unperturbed cells.

### Accumulation of Tetrameric GFP and Un-Polymerizable Tubulin in the "Nuclear Volume" at GVBD

Previous investigators suggested that tubulin dimers concentrate either by binding to something in the nuclear volume (Hayashi et al, 2012; Métivier et al, 2021) or by being excluded by cytoplasmic organelles that are kept out of the spindle volume by the ER envelope that still envelopes the spindle after NEBD (Schweizer et al, 2015, Fig 4A). We analyzed the behavior of two fluorescent probes designed to address three major issues: (1) incomplete depolymerization by nocodazole; (2) concentration of alpha/beta dimers in the absence of nocodazole; (3) specific binding of tubulin to a nuclear binding site versus volume exclusion by cytoplasmic organelles. GFP::GCN4-pLI is a tetramerized GFP designed to have a native molecular weight (128 kD) similar to an alpha/beta tubulin dimer (101 kD without tag, 128 kD with mNeonGreen tag, see Fig 7A), but which should not bind to any tubulin-specific binding sites in the nuclear volume because GCN4-pLI is a well characterized synthetic four helix bundle (Mittl et al, 2000). GFP::TBA-2(T349E) is an alpha tubulin mutant that can dimerize with beta tubulin, cannot polymerize (Johnson et al, 2011), and should bind any specific tubulin binding sites in the nuclear volume. Both GFP::GCN4-pLI (Fig 4B; Video 3) and GFP::TBA-2(T349E) (Fig 4C; Video 4) concentrated in the nuclear volume at GVBD in the absence of nocodazole. These results (1) suggested that the concentration of mNG::TBB-2 in nocodazole was not because of incomplete depolymerization; (2) that tubulin dimers concentrate in the nuclear volume during unperturbed spindle assembly; and (3) that binding to a tubulin-specific binding site in the nuclear volume is not required for concentration.

Because GFP::GCN4-pLI and GFP::TBA-2(T349E) could be tracked in the absence of nocodazole, we could examine their behavior during normal meiotic divisions. Chromosome segregation was normal in GFP::TBA-2(T349E) oocytes and GFP::GCN4-pLI oocytes (12/12 and 10/10 filmed respectively) suggesting that expression of GFP::TBA-2(T349E) or GFP::GCN4-pLI did not severely disturb normal spindle function. Interestingly, their concentration at GVBD lasted through Metaphase I but diffused to a 1:1 spindle: cytoplasm ratio at Anaphase I, followed by re-accumulation at metaphase II and dispersion at Anaphase II (Fig 4B–D; Video 4 and Video 5). This raises the question of what barrier between spindle and cytoplasmic volume might change between anaphase I and metaphase II.

### The ER Delimits the Accumulation of Free Tubulin during Meiosis and Early Mitosis

Re-accumulation of tubulin within the metaphase II spindle volume could be because of nuclear import if the nuclear envelope transiently reformed or the re-accumulation could be because of cell-cycle changes in the structure of the ER that surrounds the spindle (Fig 5A; Poteryaev et al, 2005). To test whether a functional nuclear envelope reforms transiently between anaphase I and metaphase II, we tracked GFP::NPP-6, a nuclear pore protein of the Y complex (Galy et al, 2003) (Fig 5B) and GFP::LMN-1, the *C. elegans* nuclear Lamin (Galy et al, 2003) (Fig 5C) by time-lapse imaging. Both proteins disappeared from the nuclear envelope at GVBD and did

---

interpreted from either single plane time-lapse imaging or z-stacks of 1–4 cell post-meiotic embryos. **(E)** Mean GFP::tubulin pixel intensity of the entire metaphase I spindle divided by the mean GFP::tubulin intensity adjacent to the spindle. **(F)** Mean GFP::tubulin pixel intensity of the entire metaphase I spindle divided by the mean mCherry::histone H2b intensity of the brightest half bivalent. **(G)** Anaphase I velocities measured as the increase in distance between separating chromosome masses divided by time. Bar = 5 $\mu$m. *$P < 0.05$, **$P < 0.01$.

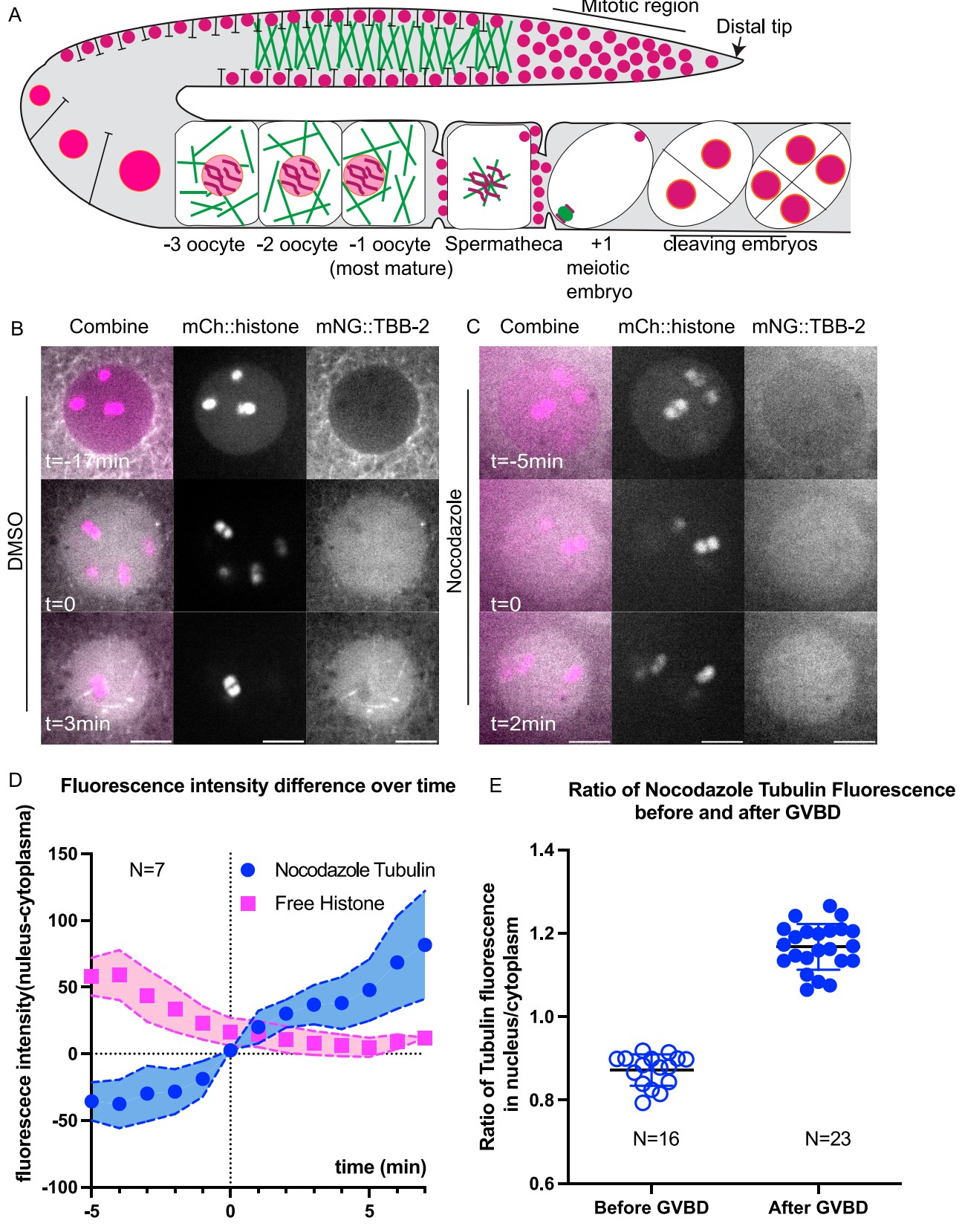

not re-locate to the nuclear membrane until pronuclear formation (Fig 5B and C). The absence of NPP-6 from a nuclear envelope during meiosis has been previously reported (Penfield et al, 2020). These results make it unlikely that tubulin concentrates in the spindle volume at metaphase II because of nuclear import. In contrast, the ER is contiguous with the outer nuclear membrane before GVBD (Fig 5A and D). After GVBD, the ER wraps around meiotic spindles with clustering at spindle poles during metaphase I and metaphase II (Fig 5D). In contrast, yolk granules (Fig 5D), maternal mitochondria (Fig 5E), and lipid droplets (Fig 5F) are excluded from the nuclear and spindle region. At the light microscope level (Fig 5D), the ER appears reticular during metaphase I and metaphase II and disperses during anaphase I and anaphase II as described previously (Poteryaev et al, 2005). To determine the ultrastructural changes in ER morphology, we manually segmented the ER in previously published electron tomograms (Lantzsch et al, 2021). During metaphase I and metaphase II, the sides of the spindle are enclosed by overlapping sheets of ER (Video 6, Fig 6A; Video 7) and the accumulation of ER at the spindle poles is a complex mixture of sheets and tubules (Fig 6B; Video 8). During anaphase I and anaphase II, the ER near the spindle consists entirely of tubules (Fig 6C and D; Video 9). By time-lapse imaging of worms expressing an ER marker as well as GFP::GCN4-pLI, we found that the dramatic morphological change of ER coincides with the concentration and dispersion of GFP::GCN4-pLI during meiosis (Video 5) and mitosis (Video 10). Thus, tubulin-sized proteins accumulate in the spindle volume when it is encased by sheet-like ER and disperse when the ER is tubular. These results favor a model in which a semi-permeable spindle envelope composed of sheet-like ER excludes a "crowding agent" from the spindle volume, so that tubulin-sized proteins concentrate in the spindle volume by volume exclusion from the surrounding cytoplasm (Fig 4A).

The "crowding agent" might consist of membranous organelles like yolk granules (Fig 5D), or mitochondria (Fig 5E) as suggested by Schweizer et al (2015) or might consist of ribosomes as suggested by Delarue et al (2018). To test whether ribosomes are a viable candidate for excluding volume from the cytoplasm, we determined the density of ribosomes inside of the spindle and outside the ER envelope from previously published electron tomograms (Lantzsch et al, 2021) and found no significant difference in ribosome density inside versus outside the spindle (Fig 6E). As a complementary approach, we monitored a ribosomal protein, GFP::RPL-29, by time-lapse imaging before and after GVBD (Fig 6F). GFP::RPL-29 was excluded from the nucleus before GVBD, then rapidly rushed into the nuclear volume at GVBD in 9/9 time-lapse sequences, indicating that ribosomes could not be the cytoplasmic crowding agent causing concentration of tubulin-sized proteins that stay concentrated through metaphase.

To test the idea that yolk granules, mitochondria, and other organelles are the crowding agent that excludes soluble tubulin from the cytoplasm, we segmented all the membranous organelles visible in previously described electron micrographs (Howe et al, 2001; Bembenek et al, 2007; Fig S4A–D). In one thin section of a −1 oocyte, membranous organelles occupied 25% of total non-nuclear area. In two different sections covering an entire metaphase II embryo, organelles occupied 21% and 22% of total area. Membranous organelles occupied 28% and 31% of the non-spindle area in these sections. The reduced cytoplasmic volume outside the nucleus or metaphase spindle would thus be between 79% and 69%. The expected apparent concentration of molecules in the nuclear/spindle volume would thus be 1/.79–1/.69 or 1.3–1.4-fold. This is reasonably close to the observed 1.2-fold enrichment given the limitations of both measurements.

Notably, GFP::RPL-29 did not concentrate in the nuclear volume at GVBD but instead equilibrated to equal fluorescence intensity inside and outside the spindle envelope. This result indicated that there is specificity to the types of molecules that concentrate in the nuclear/spindle volume.

## The Concentration of Molecules during GVBD Is Size Dependent

In immature starfish oocytes, fluorescent dextrans of 25 kD or larger are excluded from the nucleus (Lénárt et al, 2003), similar to our results with mNG::TBB-2, GFP::GCN4pLI, and GFP::TBA-2(T349E), presumably because they are too large to diffuse freely through NPCs, and/or because of nuclear export of tubulin (Schwarzerová et al, 2019). In contrast, 10 kD fluorescent dextrans accumulated in the nucleus of immature starfish oocytes at a concentration twice that of the cytoplasm (Lénárt et al, 2003). It was suggested that this is because the small dextrans diffuse freely through NPCs and because yolk granules occupy 50% of the cytoplasmic volume thus driving apparent concentration of small dextrans in the nucleus (Lénárt et al, 2003). We found that a 36 kD monomeric GFP concentrated in C. elegans oocyte nuclei before GVBD (Fig 7A and B) to a concentration twice that of the cytoplasm (Fig 7D), like 10 kD dextrans in starfish oocytes. To rule out the possibility that this might be mediated by a cryptic NLS on GFP, we expressed a 34 kD monomeric HALO tag in the C. elegans germline, which also concentrated to a twofold higher concentration in the nucleus relative to the cytoplasm in diakinesis oocytes before GVBD (Fig 7C–E).

These results suggested that the same mechanisms driving concentration of larger proteins during GVBD might be responsible for the concentration of smaller proteins before GVBD. However, monomeric HALO tag did not stay concentrated after GVBD and instead diffused to a 1:1 fluorescence ratio inside and outside the spindle envelope (Fig 7E and F). The monomeric HALO tag did not

**Figure 3. Unpolymerized tubulin concentrates in the nuclear volume at germinal vesicle breakdown (GVBD).**
**(A)** Diagram of chromosome and microtubule organization in the C. elegans gonad. DNA (magenta); Microtubules (Green); Plasma Membrane (Black); Nuclear Envelope (Orange); −1 oocyte: most mature prophase-arrested oocyte; +1 embryo: fertilized embryo undergoing meiotic divisions. **(B, C)** Representative time-lapse images of the Germinal Vesicle in −1 oocyte expressing mNG::TBB-2 (mNeonGreen::tubulin: greyscale) and mCh::histone (mCherry::histone H2b: magenta). **(B, C)** Tubulin concentrated in the nuclear volume at GVBD in worms treated with (B) DMSO or (C) Nocodazole. Non-chromosomal histone in the "nucleus" diffuses out at the onset of GVBD. bGVBD, GVBD, aGVBD: before, at and after GVBD. Scale bars, 5 μm. **(D)** Plots of fluorescence intensity difference in nucleus and cytoplasm over time after nocodazole treatment. Tubulin (blue); Histone (Magenta). Y axis: fluorescence intensity in nucleus - fluorescence intensity in cytoplasm. N: number of time lapse sequences analyzed. Mean shown in solid squares (His) or solid circles (Tub). SEM shown in colored regions. **(E)** Ratio of mean fluorescence intensity of nocodazole-tubulin in nucleus over cytoplasm before GVBD and after GVBD. Tubulin were excluded from nucleus before GVBD (ratio < 1) and concentrated in nucleus after GVBD (ratio > 1). N: number of nuclei analyzed.

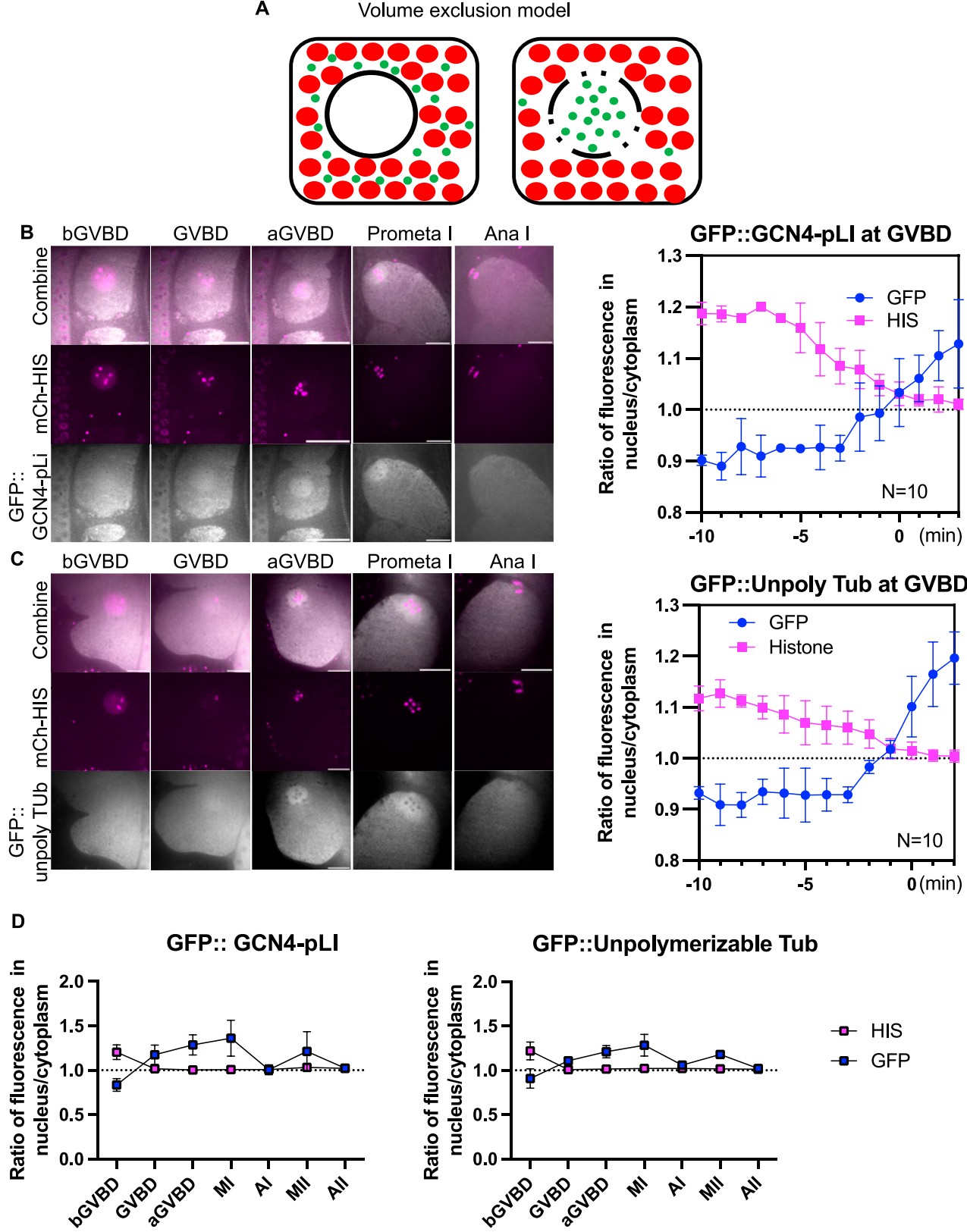

**Figure 4. Tubulin-sized molecules concentrated in the nuclear volume at germinal vesicle breakdown (GVBD).**
**(A)** Volume exclusion model of free tubulin rushing into nucleus (black circled region) where more space is available to tubulin-sized molecules (green dots) at GVBD because of volume occupied by mitochondria and yolk granules (red dots). **(B)** Representative time lapse images of meiotic embryo expressing tetrameric GFP::GCN4-pLI

grossly perturb meiotic progression (Fig 5E) in 12/12 time-lapse sequences. These results suggested that there might be a size-dependence for concentration of proteins in the nuclear volume during meiotic spindle assembly.

### Molecule Movement during GVBD Is Charge Dependent

Although the tubulin-sized molecules that concentrated in the nuclear volume at GVBD are larger than the smaller HALO tag that quickly dispersed to a 1:1 ratio between nuclear and cytoplasmic volumes at GVBD, these proteins also differ in net charge (Fig 7A), with the concentrating proteins more negative than the dispersing proteins. Single molecule diffusion studies in cytoplasm (Xiang et al, 2020) and inside organelles (Xiang et al, 2023) have revealed that net positive charge can slow or stop diffusion of small proteins. One possibility is that proteins with positive charge would interact transiently by ion exchange with negatively charged phosphatidyl serine-rich intracellular membranes, and this electrostatic interaction would slow diffusion of positively charged proteins into a fenestrated spindle envelope but would not affect diffusion of negatively charged proteins. To test this idea, we added arginines to GFP::GCN4-pLI to either neutralize net charge or add a net positive charge. Similar to GFP::GCN4-pLI (Fig 7A, net charge at pH 7.0: −28), GFP::GCN4-pLI (neutral), or GFP::GCN4-pLI (positive) (Fig 7A, net charge at pH 7.0: 0 or 56, respectively) were excluded from the nucleus before GVBD (Fig 8A–D; −10 min). GFP::GCN4-pLI (neutral) remained excluded from the nuclear volume at GVBD for a longer period of time after histone leakage out of the nucleus than GFP::GCN4-pLI and exhibited only a slight accumulation in the nuclear volume (~7 min after GVBD onset) (Fig 8A and C), a significant delay compared with GFP::GCN4-pLI (~2 min after GVBD onset, Fig 4C). GFP::GCN4-pLI (positive) remained excluded from the nuclear volume for an even longer period after GVBD (Fig 8B and D). This result suggested that proteins with net negative charge concentrate in the nuclear volume at GVBD whereas proteins with neutral or positive charges do not. Interestingly, GFP::GCN4-pLI (extra negative) (Fig 7A, net charge at PH 7.0: −56) did not accumulate in the nuclear volume as GFP::GCn4-pLI did (Fig S5), suggesting there might be a narrow range of charge that are not retained in the cytoplasm.

## Discussion

Our data suggest that the ran pathway is dispensable for spindle formation but is important for limiting meiotic anaphase velocity and polar body extrusion. This is consistent with the localization of endogenously tagged Ran-GEF, RAN-3, which dispersed from chromosomes during meiotic spindle assembly but then concentrated on chromosomes during anaphase. It remains possible that the ran pathway is redundant with another pathway, that there is chromosome-associated RAN-3 during spindle assembly that is below our detection limit, or that if RAN-3 were more completely depleted, then an assembly defect would be observed. It is also possible that other chromatin-associated activators of spindle assembly, such as the CPC (Divekar et al, 2021), katanin (McNally et al, 2006, 2014), and CLS-2 (Schlientz & Bowerman, 2020) substitute for Ran-GTP during *C. elegans* meiosis more robustly than in other species.

Increased anaphase velocities have been reported after double depletions of three microtubule crosslinkers, KLP-19, BMK-1, and SPD-1 (Li et al, 2023). Thus, Ran-GEF on anaphase chromosomes might generate Ran-GTP that would locally activate one or more microtubule crosslinkers in the anaphase spindle. RNAi depletion of RAN-3 and RAN-2 had the opposite effect, slowing anaphase, in mitotic spindles with ablated centrosomes (Nahaboo et al, 2015), suggesting different mechanisms at work during meiosis versus mitosis. Ran-GEF on anaphase chromosomes is closely juxtaposed against the cortex where Ran-GTP might activate components of the acto-myosin machinery to promote polar body formation as occurs in mouse oocytes (Deng et al, 2007).

The concentration of soluble tubulin in the nuclear volume during and after GVBD in *C. elegans* oocytes treated with noco-dazole recapitulates what has been reported in mitotic cells of *C. elegans* and *Drosophila* (Hayashi et al, 2012; Yao et al, 2012; Schweizer et al, 2015; Métivier et al, 2021). Remaining questions are the mechanism and significance of this phenomenon. Hayashi et al (2012) concluded that tubulin binds to something in the nuclear volume because of a fraction of nocodazole-treated GFP::tubulin that recovered slowly after photobleaching. Métivier et al (2021) concluded that tubulin binds to tubulin folding cofactor E in the nucleus because depletion of cofactor E abrogated concentration of nocodazole-treated GFP::tubulin. Our finding that tetrameric GFP concentrates in the nuclear volume during GVBD supports the idea that a non-specific biophysical difference between the nucleoplasm and cytoplasm is responsible. This idea is consistent with the concentration of fluorescent dextrans (carbohydrates) in the spindle volume of *Drosophila* embryos (Yao et al, 2012) and the observation by Schweizer et al (2015) that the mobility of nocodazole-treated GFP::tubulin molecules inside and outside the spindle volume are identical in S2 cells. Previous studies of nocodazole-treated GFP::tubulin could not address the cell cycle regulation of this phenomenon because nocodazole blocks spindle progression. We found that concentration of tetrameric GFP or an unpolymerizable tubulin mutant during metaphase I and metaphase II correlated with the encasement of the spindle with layered sheets of ER and the dispersal during anaphase I and II correlated with dispersed tubular ER. The idea that a semipermeable envelope around the spindle is required for the concentration of soluble

---

(greyscale) and mCh::HIS (mCherry::histone H2b: magenta). The ratio of fluorescence intensity of GFP::GCN4-pLI or non-chromosome histone in nucleus to cytoplasm during GVBD over time is shown in the graph on the right. N: number of time lapse sequences analyzed. Means are shown in solid square (His) or solid circle (Tub). Bars indicate SEM. Scale Bars, 10 μm. **(C)** Representative time lapse images of a meiotic embryo expressing un-polymerizable GFP::TBA-2(T349E) (greyscale) and mCh::histone (magenta). The ratio of fluorescence intensity of GFP or non-chromosome histone in nucleus to cytoplasm during GVBD over time is shown in the graph on the right. Scale Bars, 10 μm. **(D)** Plots of fluorescence intensity ratio in the nucleus or spindle to cytoplasm before germinal vesicle breakdown (bGVBD), GVBD, or after GVBD (aGVBD), metaphase I (MI), anaphase I (AI), metaphase II (MII) and anaphase II (AII) show concentration at MI and MI, and dispersion at AI and AII.

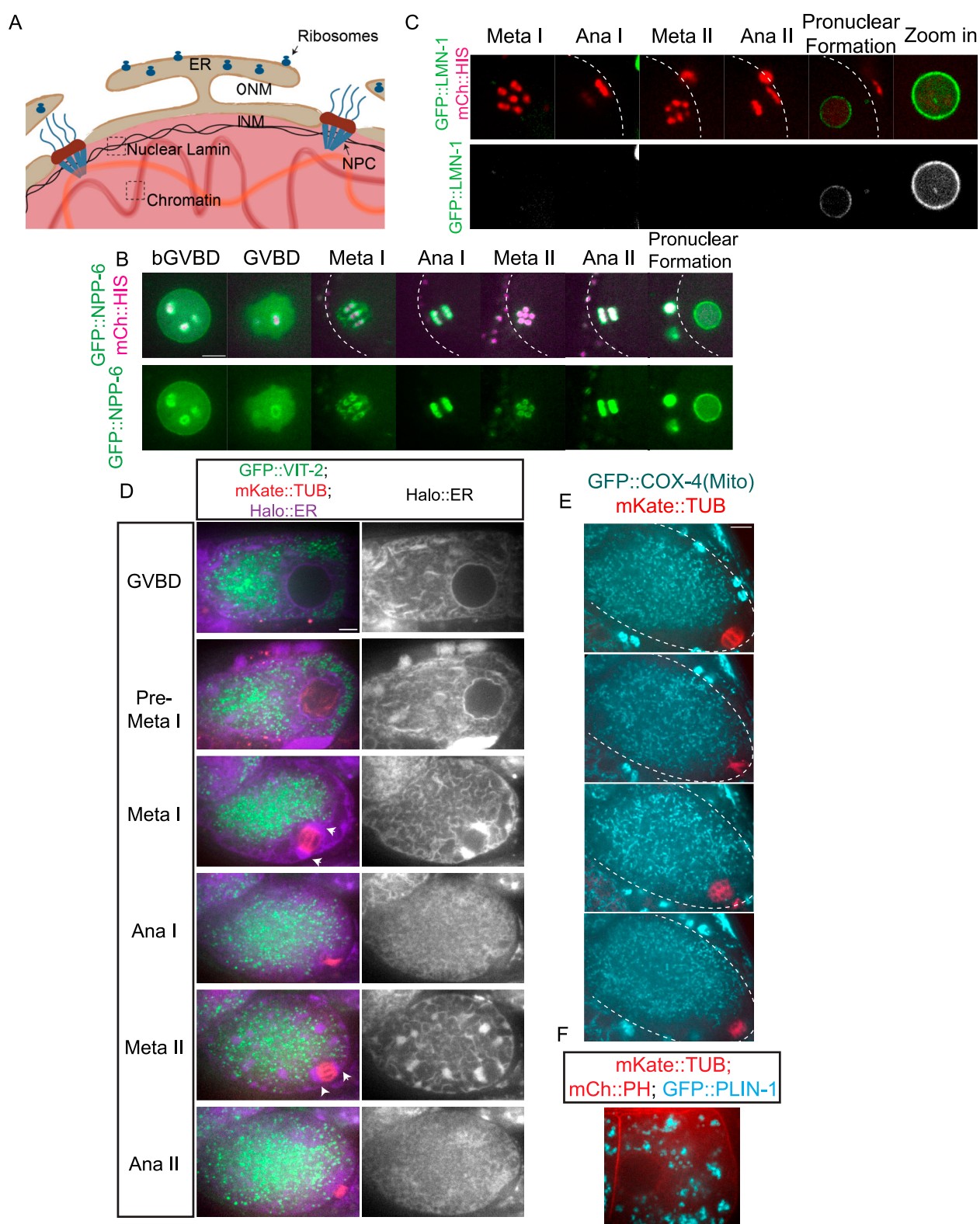

**Figure 5. The ER envelope delimits the concentration of free tubulin during meiosis.**
**(A)** Diagram of an intact nucleus before GVBD. ONM: Outer Nuclear Membrane. INM: Inner Nuclear Membrane. NPC: Nuclear Pore Complex. ER: Endoplasmic Reticulum.
**(B, C, D, E, F)** Time-lapse sequences of representative meiotic embryo expressing GFP::NPP-6 (green); mCh::histone (red) or (C) GFP::LMN-1 (green); mCh::histone (magenta) or (D) GFP::VIT-2 (yolk granule cargo: green); mKate::TUB (red); HALO::ER (magenta) or (E) GFP::COX-4 (mitochondrial protein: cyan); mKate::TUB (red) or (F) GFP::PLIN-1 (lipid droplet protein: cyan); mCh::PH (red); mKate::TUB (red). Scale Bars = 5 $\mu$m. The cell cortex was drawn in white dashed line in (B, C, E).

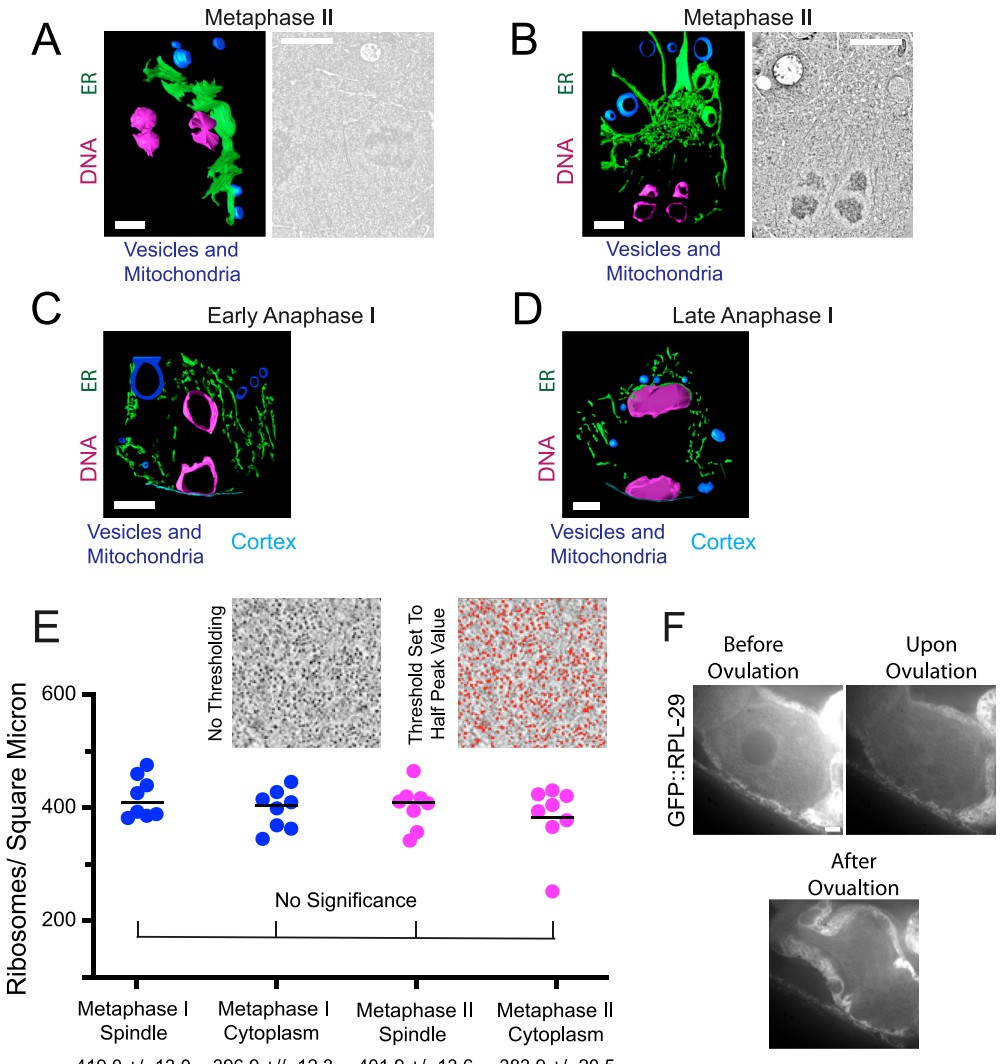

**Figure 6. ER sheets envelop metaphase meiotic spindles.**

**(A)** Left. Model of ER sheets on the partial exterior of a metaphase II meiotic spindle derived from an electron tomogram and spanning 1.2 $\mu m$ in the z. Right. Single plane image from the tomogram. The two lobes of each univalent chromosome are oriented down the pole to pole axis of the spindle. **(B)** Left. Model of ER at one pole of a metaphase II meiotic spindle derived from an electron tomogram and spanning 0.6 $\mu m$ in the z. Right. Single plane image from the tomogram. The two lobes of each univalent chromosome are oriented down the pole to pole axis of the spindle. **(C)** Model of ER in an outer 0.6 $\mu m$ z-section of a MI early anaphase spindle. No large ER sheets were observed. **(D)** Model of ER in a 0.6 $\mu m$ z-section of a late anaphase I spindle. **(A, B, C, D)** ER in green, chromosomes in magenta, cytoplasmic organelles in blue. **(A, B, C, D)** Scale bar, 1 $\mu m$. **(E)** Ribosomes were counted in representative squares in representative sections of spindle and cytoplasm in metaphase I and metaphase II electron tomograms. Ribosomes were counted after setting the threshold value to half of the peak value. Average ribosome areas in metaphase I spindles and cytoplasm, and metaphase II spindles and cytoplasm were: 30.9 ± 2.0 nm, 28.0 ± 1.2 nm, 26.3 ± 1.0 nm, and 26.8 ± 1.7 nm respectively. **(F)** Time-lapse images of meiotic embryo expressing GFP::RPL-29 (grayscale) during GVBD. Scale Bar, 5 $\mu m$.

tubulin was supported by Schweizer et al (2015) who showed that laser cutting of the spindle envelope prevents the concentration of soluble tubulin. Genetic or physical perturbation of the ER in *C. elegans* could help address both the mechanism of tubulin concentration and the significance. We expect that any perturbation that disrupts the ER spindle envelope would prevent concentration of free tubulin and cause a delay in spindle assembly. If pathways are redundant, severe spindle assembly defects might only be observed after combining depletions with other chromosomal SAFs like RAN-3, katanin, or the CPC.

Our results indicate that both size and net charge determine whether a protein will concentrate in the nuclear volume before and after GVBD. We suggest that sheet-like ER is required to exclude mitochondria, yolk granules, and other organelles from the spindle volume. The cytoplasmic faces of cytoplasmic organelles are thought to be negatively charged because of phosphoinositides and phosphatidylserine. A recent study mapping the electrostatic profile of cellular membranes suggests that plasma membrane, ER, mitochondria, Golgi are all negatively charged in Hela cells (Eisenberg

et al, 2021; average surface potential ranging from –14 mV to –35 mV). The negatively charged lipid heads will have counterions neutralizing these charges. However, it is possible the negative surface charge of cellular membranes might act like cation exchange chromatography beads, causing transient binding of proteins with net positive charges. In this scenario, diffusion of positively charged proteins into the spindle volume would be slowed, whereas diffusion out to the cytoplasm would be unrestricted. In contrast, proteins with a strong net negative charge would diffuse freely in both directions but appear concentrated in the spindle volume because of the low available volume between organelles in the cytoplasm. One logical outcome of this model is that tubulin concentration at the nanoscale between organelles would actually be no different than in the nuclear/spindle volume. However, the total mass of microtubules that could polymerize would be severely restricted, especially if a polymerizing plus end was stimulated to catastrophe by collision with an organelle. These ideas could readily be tested with in vitro experiments. A recent study found that a katanin mutant that cannot bind microtubules still concentrates in the spindle volume

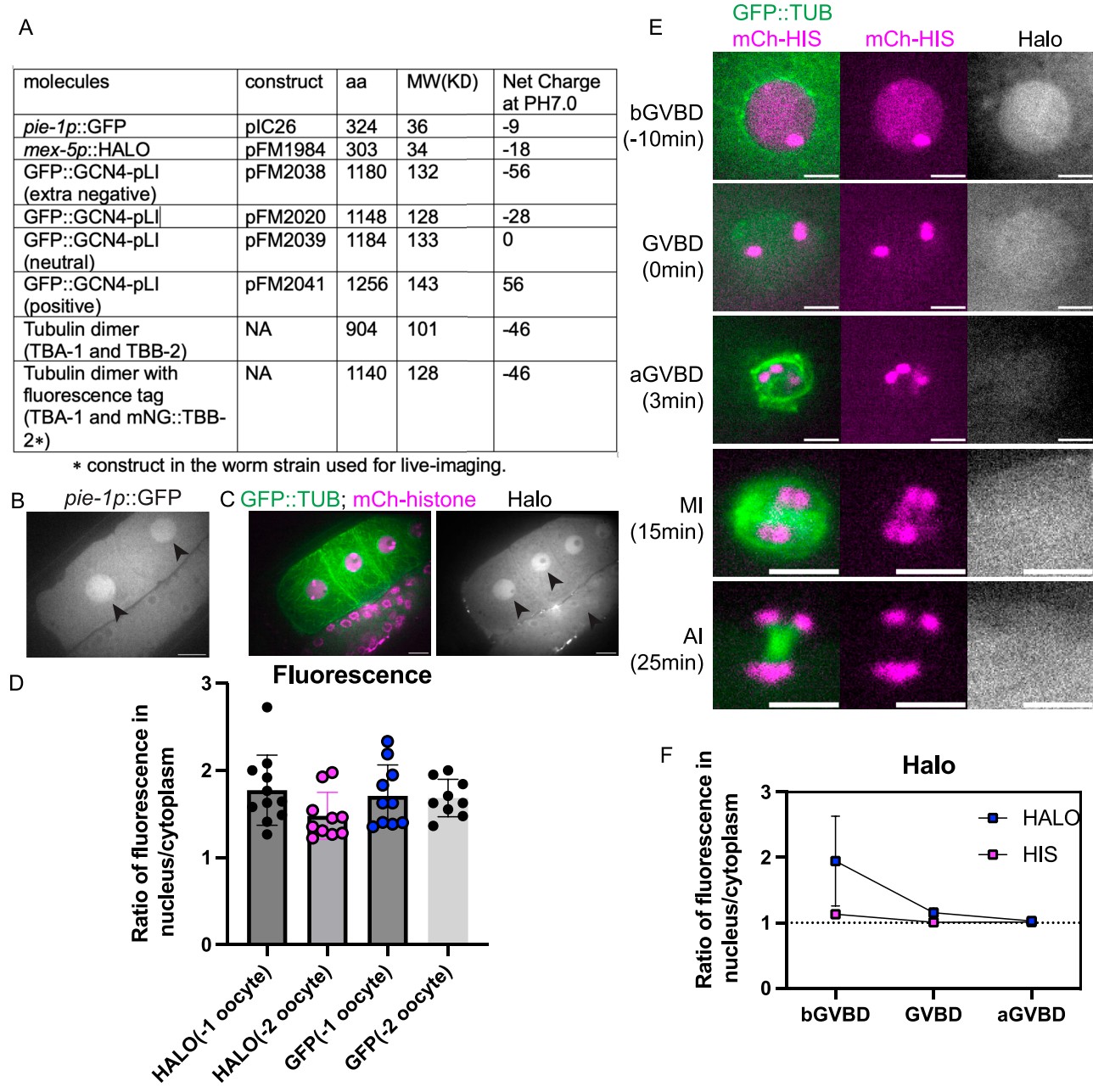

**Figure 7. Molecule movement during GVBD is size dependent.**
**(A)** Molecular weight and net charge of molecules used in this study, expressed in the *C. elegans* germline. **(B, C)** Images of diakinesis oocytes (B) expressing GFP (greyscale) before Germinal Vesicle Breakdown, and (C) expressing GFP::TUB (green), mCh::histone (magenta) and HALO (greyscale). Scale Bars, 10 $\mu$m. **(D)** Fluorescence intensity ratio of GFP or HALO in the nucleus to the cytoplasm in −1 or −2 oocytes. **(E)** Time lapse images of meiotic embryo expressing GFP::TUB (green), mCh::histone (magenta) and HALO (greyscale). Scale Bars, 5 $\mu$m. **(F)** Fluorescence intensity ratio of HALO and non-chromosome histone in the nucleus to the cytoplasm before GVBD, at GVBD onset and after GVBD.

(Beaumale et al, 2024) suggesting that other SAFs besides tubulin may be concentrated by the same biophysical mechanism.

# Materials and Methods

## *C. Elegans* Strains

*C. elegans* strains used in this study are listed in Table S1.

## Transgenes Generated for This Study

syb7781 and syb7819 were generated by inserting Auxin Induced Degron (AID) and HALO tag sequences to the endogenous *ran-3* and *ran-2* loci through CRISPR/Cas9-mediated genome editing by Suny Biotech. Sequences of these insertions are in Supplemental Data 1. *duSi18*, *duSi20*, *duSi23*, *duSi24*, and *duSi25* were generated by inserting GFP::GCN4-pLI (or its charge variants), or GFP::tba-2(T349E) via Flp Recombinase-Mediated Cassette Exchange (RMCE) method

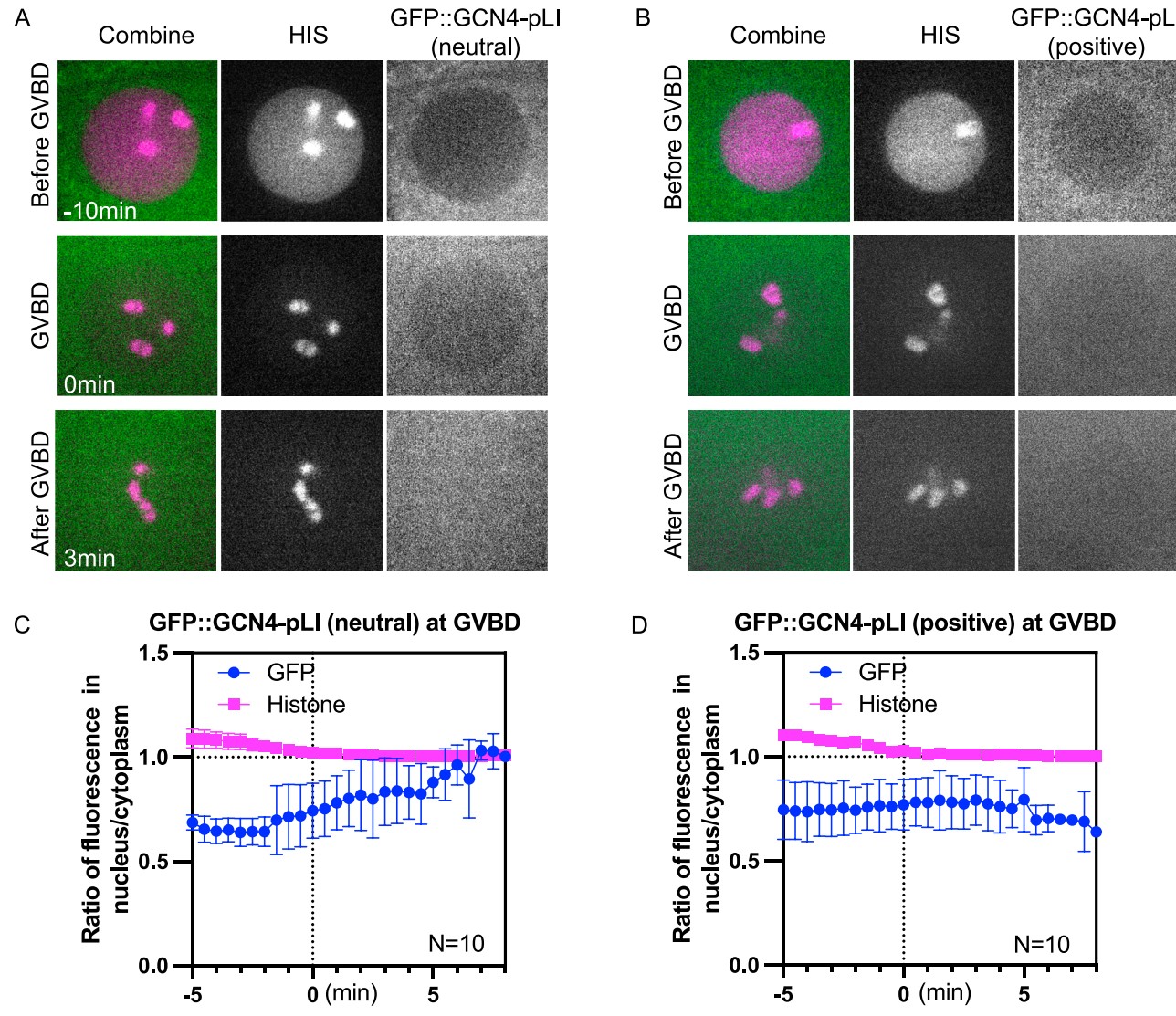

**Figure 8. Molecule movement during GVBD is charge dependent.**
**(A, B)** Representative time lapse images of −1 oocyte expressing mCh::histone and GFP::GCN4-pLI (neutral) or (B) GFP::GCN4-pLI (positive). **(C, D)** Plots of fluorescence intensity ratio in nucleus to cytoplasm over time. Y axis: fluorescence intensity (nucleus-background) ÷ fluorescence intensity (cytoplasm-background). N: number of time lapse sequences analyzed. Mean is shown in solid magenta square (His) or solid green circle (GFP). Bars indicate SEM.

(Nonet, 2020). The plasmids containing corresponding DNAs (GFP::GCN4-pLI, GFP::GCN4-pLI [extra negative], GFP::GCN4-pLI [neutral], GFP::GCN4-pLI [positive], GFP::tba-2[T349E]) were injected into *jsSi1579* landing site in NM5402 by In Vivo Biosystems, then the NeonGreen + FLP transgene was removed by outcrossing. Plasmids for RMCE insertion are constructed as follows: DNA sequences of GCN4-pLI and tba-2(T349E) were obtained as gBlocks from IDT followed by restriction digestion and cloned into the vector by T4 DNA ligation. The vector is designed for RMCE insertion, and it contains GFP optimized to reduce germline silencing. For charge variants of GFP::GCN4-pLI, gblocks containing aspartic acid and glutamic acid were cloned into GFP::GCN4-pLI to add negative net charge. To neutralize negative charge of GFP::GCN4-pLI or add positive net charge, 7 arginine or 21 arginine were cloned onto GFP::GCN4-pLI. The strains were sequenced, and the sequences are listed in Supplemental Data 2.

**Drug Treatment**

The strain used for nocodazole experiments has a *bus-17(e2800)* mutation which makes the worms cuticle permeable to drugs. 5 mg/ml stock nocodazole solution (Sigma-Aldrich, dissolved in 100% DMSO) was diluted into tricaine/tetramisole anesthetics to 5 µg/ml just before adding to worms for live imaging. 100% DMSO without nocodazole was diluted in the same way for control treatment.

**Live Imaging and Statistical Analysis**

Worms were anesthetized with tricaine/tetramisole as described (Kirby et al, 1990; McCarter et al, 1999) and gently mounted between a coverslip and a thin 2% agarose pad on a slide. All

time lapse images were captured with a Solamere spinning disk confocal microscope equipped with an Olympus IX-70 stand, Yokogawa CSU10, either Hamamatsu ORCA FLASH 4.0 CMOS (complementary metal oxide semiconductor) detector or Hamamatsu ORCA-Quest qCMOS (quantitative complementary metal oxide semiconductor) detector, Olympus 100x UPlanApo 1.35 objective, 100-mW Coherent Obis laser set at 30% power, and MicroManager software control. Pixel size was 65 nm for the ORCA FLASH 4.0 CMOS detector and 46 nm for the ORCA-Quest qCMOS detector. For measuring fluorescence intensity or nucleus sizes (in Figs 2A–C and 7B and C and S1A), z-stack images (except in Fig S2A) were taken on the same microscope, with 1-micron step size to capture the center of nucleus for measurements. Images in Fig S3A were captured with a Zeiss Laser Scanning microscope (LSM) 980, Zeiss Objective LD LCI Plan-Apochromat 40x/1.2 Imm Corr DIC M27 for water, silicon oil or glycerin. For time-lapse movies of GVBD and meiosis, images were captured every 30 s. The fluorescence intensity in the background (where there was no worm) was subtracted before the fluorescence in the nucleus or cytoplasm before quantification.

For segmentation of ER in electron tomograms, ER structures were manually assigned at 10 nm intervals using IMOD software (https://bio3d.colorado.edu/imod/). Ribosomes in 1 $\mu m^2$ sections of spindle and cytoplasm in both metaphase I and metaphase II tomograms were counted using ImageJ software. Threshold values for each section were set to half of the peak value and the number of particles was determined. The average particle size in metaphase I spindles and cytoplasm, and metaphase II spindles and cytoplasm was: 30.9 ± 2.0 nm, 28.0 ± 1.2 nm, 26.3 ± 1.0 nm, and 26.8 ± 1.7 nm, respectively. Anaphase velocities were reported as the increase in distance between chromosome masses divided by time.

Area occupied by membranous organelles was determined from previously reported electron micrographs (Howe et al, 2001; Bembenek et al, 2007). Organelles were manually segmented using the Labkit plugin in Image J. Resulting segmented layers were exported as bitmap tifs and total area was determined with the Measure function in Image J. Numerical values corresponding to data points in figures are included in Supplemental Data 3.

# Supplementary Information

# Acknowledgements

We thank Kent McDonald, Barbara Meyer, and Josh Bembenek for sharing electron microscopy data and Shuyan Qui for generating the PLIN-1 strain. We thank Ho-yi Mak, Jim Priess, Arshad Desai, and the Caenorhabditis Genetics Center (CGC) for providing strains. The CGC is funded by the NIH Office of Research Infrastructure Programs (P40 OD010440). This work was funded by National Institute of General Medical Sciences grant R35GM136241 to FJ McNally and by the US Department of Agriculture/National Institute of Food and Agriculture Hatch project (1009162 to FJ McNally).

## Author Contributions

T Gong: conceptualization, data curation, investigation, and writing—original draft.
KL McNally: data curation, formal analysis, investigation, and writing—review and editing.
S Konanoor: resources, investigation, and methodology.
A Peraza: resources, investigation, and methodology.
C Bailey: resources, investigation, and methodology.
S Redemann: resources, investigation, and methodology.
FJ McNally: conceptualization, data curation, formal analysis, funding acquisition, investigation, and writing—original draft, review, and editing.

## Conflict of Interest Statement

The authors declare that they have no conflict of interest.

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
