## [Reviewer comments · Life Science Alliance]

Life Science Alliance

Roles of Tubulin Concentration during Prometaphase and Ran-GTP during Anaphase of *C. elegans* meiosis

Ting Gong, Karen McNally, Siri Konanoor, Alma Peraza, Cynthia Bailey, Stefanie Redemann, and Francis McNally
DOI: <https://doi.org/10.26508/lsa.202402884>

Corresponding author(s): Francis McNally, University of California, Davis

Review Timeline:

Submission Date:	2024-06-12
Editorial Decision:	2024-06-13
Revision Received:	2024-06-20
Editorial Decision:	2024-06-25
Revision Received:	2024-06-25
Accepted:	2024-06-26

Transaction Report:

Please note that the manuscript was previously reviewed at another journal and the reports were taken into account in the decision-making process at *Life Science Alliance*.

Reviews

Reviewer #1 Review

Comments to the Authors (Required):

Gong et al present evidence that the Ran small GTPase does not participate in early oocyte meiotic spindle assembly in *C. elegans* but instead has only a modest role during anaphase, when as the authors show both RAN-2 and -3 are enriched on chromatin and largely undetectable around the forming spindle earlier in meiotic cell division. The authors suggest that an alternative mechanism for the early nucleation of spindle microtubules is the concentration of tubulin in the nucleus after nuclear envelope breakdown begins, due to volume exclusion outside of the nucleus by other cellular components and to endoplasmic reticulum sheets excluding cellular organelles from the spindle area during metaphase I and II.

While these results are interesting and propose a novel mechanisms for microtubule nucleation during *C. elegans* oocyte meiosis, the manuscript is exceptionally lacking in conclusiveness. The lack of a requirement for Ran earlier in spindle assembly may be due to insufficient depletion, although the effort to avoid defects due to nuclear import/export problems is admirable. Moreover, the authors have not sufficiently quantified their data on this point (see below). Furthermore, while the data supporting enrichment of tubulin around chromatin being due to volume exclusion is compelling, and their analysis of that process interesting, the authors provide no evidence that altering tubulin concentration in the nucleus impacts spindle assembly. Similarly, while the data on the ER is interesting it is only descriptive and no evidence is provided that alterations in ER distribution/structure impact spindle assembly. Given the lack of conclusiveness on all key points, the manuscript is interesting but more suitable for publication in more specialized journals such as *Molecular Biology of the Cell*, the *Journal of Cell Science* and *PLOS One*.

The authors also need to address the following concerns before the manuscript can be considered suitable for publication.

1. The authors state that they detect no defects in early spindle assembly after Ran depletion. However, the RAN-3 depletion images shown in Figure 2B show a clear decrease in microtubule signal relative to the control and to the RAN-2 depletion, and the authors provide no quantification of microtubule levels. The authors need to provide quantification of integrated pixel intensity and statistics to better assess requirements for Ran during oocyte meiotic spindle assembly.

2. The authors state that Ran depletion leads to increased anaphase I velocity, and document this in Figure 2E. However, the authors never define anaphase velocity. Presumably it is the rate of chromosome separation, but this should be made clear. More significantly, the authors provide no discussion of how Ran might be participating in anaphase and why loss of Ran would lead to increased velocity. The authors should provide a more detailed discussion of why *C. elegans* might differ from other model systems with respect to a requirement for Ran and how it might be participating in anaphase.

Reviewer #2 Review

Comments to the Authors (Required):

In the manuscript, the authors aim to find out the mechanism of initiation of spindle assembly in *C. elegans* oocytes. Spindle assembly in oocytes takes place without centrosomes in many species including humans. However, how spindle assembly is initiated in oocytes still remains unanswered. Dispensability of Ran-GEF (Rcc1) has been shown in other systems using a dominant negative mutation, but this study used a degron system to show Ran-GEF is indeed dispensable for spindle assembly. in *C. elegans* oocytes They looked for an alternative mechanism for spindle assembly. They found a higher concentration of tubulin dimers in the nuclear regions during nuclear envelope breakdown, and investigated the mechanism of tubulin accumulation. They found a correlation between ER forming sheets and exclusion of cytoplasmic organelles from the spindle regions, and also that molecular sizes and charges of protein complexes appear to determine whether they are accumulated in the nuclear/spindle region.

Various hypotheses on the mechanisms of tubulin accumulation are systematically tested using various neat experiments or analysis. These experiments are well executed and the findings are themselves worth publishing, but a link to the initiation of spindle assembly is only a theoretical possibility. At the moment I am not sure whether the title "Mechanisms of meiotic spindle initiation in *Caenorhabditis elegans* oocytes" can be justified. No evidence is presented to support that the tubulin accumulation in the nuclear/spindle region can explain the initiation of spindle assembly. As the difference in the concentration between the nuclear/spindle region and the cytoplasm is modest, it is unclear whether or how much this actually contributes to the initiation of spindle assembly or under which circumstances this could make a significant contribution.

Inclusion of some evidence, even if it is circumstantial, would greatly improve the manuscript. Just for example, they may test whether initiation of spindle assembly is sensitive to moderate changes of the tubulin concentration (with or

without Ran-GEF), or whether disrupting the ER integrity or organisation could affect tubulin accumulation and/or spindle initiation. The later experiment could also confirm their unproven hypothesis that ER sheets surrounding the spindle exclude the cytoplasmic organelles and concentrate tubulin in the nuclear/spindle region.

p8 L18, Fig1C: "RAN-3 only faintly associated with chromosomes at metaphase I and II." The line measurements are missing at some stages, including metaphase I and II. Their inclusion is important for supporting the statement.

Fig2E. The velocity of anaphase I: this needs a control without auxin, as the authors suspect that genetic variation between the lines may cause different frequencies of polar body extrusion failure. If the velocity difference is due to depletion of RAN-2 or RAN-3, speculation on what this mean should be included.

Reviewer #3 Review

Comments to the Authors (Required):

Gong et al. addresses the well-studied question of centrosome-independent spindle assembly, using meiotic oocytes in *C. elegans* as a model. They employ new tools using the AID system to rigorously show that RanGTP is not required for *C. elegans* meiotic spindle assembly. They find that in *C. elegans* meiotic oocytes, similar to mitotic embryos as shown in Hayashi et al. 2012, soluble tubulin accumulates in the nuclear and spindle region after NEBD. The authors claim that this accumulation is not because of localized assembly of short MTs nor nuclear transport mechanisms. They cleverly use a tetrameric GFP construct similar in size to *a/b*-tubulin dimers to show that there is a size and charge dependence for accumulation of factors in the spindle region. This result together with their observation that large ER sheets surround the meiotic spindle at metaphase may provide an explanation for the concentration of negatively charged *a/b*-tubulin in this region.

The authors use sophisticated imaging and use of tools to address the question of whether a spindle-envelope concentrates tubulin to promote spindle assembly, which as cited had been proposed in PMC4555823. They make a number of interesting observations and correlations and put forth several ideas for how and why tubulin concentrates in this region (ER structure, charge and size of the molecule, size exclusion, charge of membranes etc). However, without testing these ideas and a demonstration of whether their observations are functionally important for meiotic spindle assembly, the paper does not make a significant advance in our understanding of non-centrosomal spindle assembly mechanisms.

Major issues

1. A more direct test of their model would be to prevent tubulin accumulation in the spindle region by disrupting ER morphology (introducing a genetic background that makes the ER more tubular-like or developing a system to tether ER sheets to the plasma membrane, for example).
2. A major part of the results is to demonstrate that the Ran gradient is not important for spindle assembly in oocyte meiosis in *C. elegans*. While the tools are elegant and robust, this has already been shown using RNAi and does not substantially advance understanding.
3. The paper starts out about spindle assembly, but spends a significant portion talking about the role of RanGTP in polar body extrusion, which seems unrelated to the overall question of the paper.
4. The authors use the GFP::*GCN4*-pLI probe throughout the paper but there do not clearly describe how this constructs results in tetramerized GFP. How certain are the authors that it is working as it should in *C. elegans*? It would also be helpful to directly discuss the molecular weights of the different constructs to clarify the rationale to the reader earlier in the paper.
5. Figure 5: It has been shown that GFP::*NPP_6* does not accumulate at the NE in meiosis I (PMCID: PMC7199858) so this finding is not new.
6. There is no mention of tubulins export signal throughout the manuscript. See PMCID: PMC6451007, which should be cited. Other accounts of tubulin export should also be referenced. (eg. pg. 18 lines 1-4)
7. Line 8 on pg. 19 suggests that phosphatidylserine rich intracellular membranes may be involved in the charge dependent concentration of factors. However, the authors do not test this potential mechanism (by disrupting phosphatidylserine production, for example).

Minor points

1. Since most of the study is on RanGTP gradient around chromosomes for spindle assembly, figure 1 A could focus

on this rather than traditional import/export across the NE.

2. A schematic would be useful in Fig. 2

3. While important to understand the effectiveness of the AID system, the text on pg 9 could be condensed so that there is more information about the results from the depletion rather than the use of the tool itself.

4. On line 16 the authors write "surprisingly" but it is unclear why the accumulation of tubulin in the nuclear region at this time point is surprising given that several papers in many systems that they cite show this.

5. Figure 6 and associated videos do not include the tomogram sections, only the 3D models. data. It would be better if the authors overlaid the tracings with the data. Without the EM tomograms, it's not possible to know if all of the ER or part of the ER in those sections were traced. Also, the authors should provide schematics associated with the 3D renderings so readers can understand the orientation.

6. In figure 6, the authors should provide schematics associated with the 3D renderings so readers can understand the orientation.

June 13, 2024

Re: Life Science Alliance manuscript #LSA-2024-02884-T

Dr Francis J McNally
Davis, Univ. of California at
Dept of Mol. & Cell Biology
3225 Life Sciences Addition
Davis, USA-Davis CA 95616 95616

Dear Dr. McNally,

Thank you for submitting your manuscript entitled "Mechanisms of Meiotic Spindle Initiation in *Caenorhabditis elegans* Oocytes" to Life Science Alliance. We invite you to submit a revised manuscript addressing the Reviewer comments.

Thank you for this interesting contribution to Life Science Alliance. We are looking forward to receiving your revised manuscript.

Sincerely,

B. MANUSCRIPT ORGANIZATION AND FORMATTING:

June 25, 2024

RE: Life Science Alliance Manuscript #LSA-2024-02884-TR

Prof. Francis J McNally
University of California, Davis
Dept of Mol. & Cell Biology
University of California, Davis
3167 Green Hall
Davis, USA-Davis CA 95616 95616

Dear Dr. McNally,

Thank you for submitting your revised manuscript entitled "Ran drives polar body formation and tubulin concentrates during *C. elegans* meiotic spindle assembly". We would be happy to publish your paper in Life Science Alliance pending final revisions necessary to meet our formatting guidelines.

- please be sure that the authorship listing and order is correct
- titles in the system and manuscript file should match
- please add an Author Contributions section to your main manuscript text
- please use the [10 author names et al.] format in your references (i.e., limit the author names to the first 10)
- please add a Conflict of Interest statement to your main manuscript text
- please add callouts for Figures S1F, S3C-D and S4A-D to your main manuscript text

A. FINAL FILES:

B. MANUSCRIPT ORGANIZATION AND FORMATTING:

Thank you for your attention to these final processing requirements. Please revise and format the manuscript and upload materials within 5 days.

Sincerely,

June 26, 2024

RE: Life Science Alliance Manuscript #LSA-2024-02884-TRR

Prof. Francis J McNally
University of California, Davis
Dept of Mol. & Cell Biology
University of California, Davis
3167 Green Hall
Davis, USA-Davis CA 95616 95616

Dear Dr. McNally,

Thank you for submitting your Research Article entitled "Roles of Tubulin Concentration during Prometaphase and Ran-GTP during Anaphase of *C. elegans* meiosis". It is a pleasure to let you know that your manuscript is now accepted for publication in Life Science Alliance. Congratulations on this interesting work.

DISTRIBUTION OF MATERIALS:

Again, congratulations on a very nice paper. I hope you found the review process to be constructive and are pleased with how the manuscript was handled editorially. We look forward to future exciting submissions from your lab.

Sincerely,
